# Relationship Between Human Papilloma Virus and Upper Gastrointestinal Cancers

**DOI:** 10.3390/v17030367

**Published:** 2025-03-04

**Authors:** Ömer Vefik Özozan, Hikmet Pehlevan-Özel, Veli Vural, Tolga Dinç

**Affiliations:** 1Ömer Özozan Private Clinic, Kadıköy, 34744 İstanbul, Türkiye; 2Department of General Surgery, Ankara Bilkent City Hospital, 06800 Ankara, Türkiye; hikmet.pehlevan@gmail.com (H.P.-Ö.); tolga_dr@hotmail.com (T.D.); 3Department of General Surgery, Akdeniz University, 07070 Antalya, Türkiye; velivural1980@hotmail.com

**Keywords:** HPV, human papillomavirus, esophagus cancer, gastric cancer

## Abstract

The human papillomavirus (HPV) is an oncogenic DNA virus that is the most commonly transmitted sexually transmitted virus. There is substantial evidence that HPV is associated with different types of cancer. While the majority of studies have concentrated on urogenital system cancers and head and neck cancers, the relationship between HPV and gastrointestinal system cancers, particularly esophageal cancers, has also been the subject of investigation. Given that HPV is a disease that can be prevented through vaccination and treated with antiviral agents, identifying the types of cancers associated with the pathogen may inform the treatment of these cancers. This comprehensive review examines the relationship between HPV and cancers of the upper gastrointestinal tract, highlighting the oncogenic mechanisms of the virus and its reported prevalence. A deeper understanding of HPV’s association with cancer is relevant to the further development of cancer therapies.

## 1. Introduction

Infectious diseases are involved in the etiology of more than 50% of human cancers, and about 10% are caused by viral infection [1]. Cancer-inducing viruses can lead to cancer development by encoding several proteins in the target cell, reprogramming signals in the development, growth and apoptosis stages, or by the progression of inflammation caused by chronic viral infection [2]. Human papilloma virus (HPV) is one of the viruses known to cause cancer. HPV is one of the most common sexually transmitted viral infections worldwide, affecting both men and women. Among the most important cancer-causing infectious agents worldwide, Helicobacter pylori ranked first with 36.3% and HPV ranked second with 31.1% [3]. HPV has been identified as a causative agent of various types of infections, including those affecting the skin and mucosal surfaces. While it is predominantly associated with diseases in the genital area, there is growing evidence suggesting its role in infections and cancers affecting the head and neck region, as well as the gastrointestinal system. Recent years have seen an escalating focus on the link between HPV and upper gastrointestinal tract cancers, underscoring the need for further research and clinical monitoring in this area. HPVs infect stratified epithelium and can cause cervical (>90% associated), anal, vulvar, vaginal and vulvar anogenital carcinoma and are also causally associated with mucosal epithelial carcinomas, particularly oropharyngeal (4% association) [4]. Since HPV is a common type of oncogenic virus and can be silently present in the body for many years, its oncogenic properties, treatments and preventive methods such as vaccination are frequently the subject of research. In addition, due to its popularity and common occurrence, its relationship with other types of cancer has also been investigated. Studies have shown that HPV prevalence is associated with anal, colorectal, oral, pharyngeal and esophageal carcinogenesis [5]. The association of HPV prevalence with gastric carcinogenesis is controversial, and no study has shown an association with the duodenum. The aim of this study is to review studies on the relationship between HPV and esophageal, gastric and duodenal cancers.

## 2. HPV Virus

HPV is a DNA virus that does not encode for any enzyme or polymerase, requires a host cell for replication, can only infect actively dividing cells, and targets cutaneous or mucosal squamous epithelia [6]. The HPV genome has a circular DNA structure. There are three categories of genes: early genes (nonstructural proteins; E1 to E7), which facilitate viral genome expression and replication while also regulating host cell proliferation and differentiation; late genes (structural proteins; L1 and L2), which are involved in viral capsid formation; and transcriptional control region (long control region; LCR) [7,8]. There are more than 400 types of HPV [9]. HPV is divided into five groups: gamma, beta, mu, gamma and alpha; these are classified as cutaneous or mucosal. HPV8 (beta-cutaneous) and HPV16–18–31–33–35–39–45–51–52–52–5–5–58–59 (alpha-mucosal) subtypes are at high risk for invasive cancer, particularly cervical, anogenital, and oropharyngeal cancers [10]. Some cutaneous HPV types, such as HPV-8 (beta-cutaneous), have also been linked to non-melanoma skin cancers, particularly in immunocompromised individuals [9,10]. HPV is known to be transmitted through direct contact and through certain objects. Mucosal HPV is mainly transmitted by sexual contact, but vertical transmission from mother to child in the womb and more often perinatally is possible. HPV reaches the basal layer through a wound or microdamage in the epithelium, where it can infect only the dividing keratinocytes of the basal layer.

## 3. HPV Life Cycle

Human papillomaviruses (HPVs) are small, non-enveloped, double-stranded DNA viruses that infect epithelial cells and establish long-term infections without causing immediate cytopathic effects [11]. HPV transmission occurs primarily through direct contact, and infection is facilitated by microabrasions on the epithelial surface, allowing the virus to reach basal keratinocytes [12]. Once in a basal cell, HPV maintains its genome as an episome and replicates simultaneously with host DNA during cell division [11]. This episomal state allows the virus to persist in the host without triggering an immediate immune response.

HPV has an affinity for the squamocolumnar junctional tissue because basal cells are particularly accessible in the squamocolumnar transformation zone and are particularly susceptible to viral infection [13]. Progressive acid damage to the esophagus may increase the likelihood of mucosal breaks that allow virus entry into the basal layer of the transformation zone [13]. The SCJ is the transformation zone of the esophagus and resembles the transition zone of the uterine cervix, where nearly all high-grade cervical lesions and cancers arise [13]. The presence of HPV in these areas alone is not sufficient for cancer development; cancer development is multifactorial.

HPV exhibits a tightly regulated life cycle that depends on the differentiation of infected keratinocytes [14]. In basal cells, HPV replication is restricted, maintaining a low copy number of viral genomes, typically between 50 and 100 copies per cell, and viral oncogene expression is tightly controlled [11]. As infected basal cells differentiate and migrate toward the epithelial surface, the virus enters the productive phase. During this phase, viral genome replication increases to thousands of copies per cell, and structural proteins such as L1 and L2 are expressed, enabling the assembly of infectious virions [12]. The entire life cycle from initial infection to viral particle release takes approximately three weeks [11].

A key feature of the HPV life cycle is its ability to evade immune detection by restricting viral gene expression to differentiating keratinocytes that are not actively involved in immune surveillance [14]. Unlike many other viruses, HPV does not cause cell lysis; instead, virions are released via natural epithelial shedding, allowing transmission to continue without eliciting a strong immune response [11]. HPVs are often found integrated into host DNA in premalignant lesions and cancers, but this is not part of the viral life cycle. In fact, integration is the end for the virus because it can no longer form a circular genome [14].

HPVs are frequently detected to be integrated into host DNA in premalignant and malignant lesions, but integration is not part of the normal viral life cycle [14]. In fact, integration represents a dead end for the virus as it disrupts the circular structure of the viral genome, preventing further episomal replication and transmission [15]. However, integration is a crucial event in HPV-associated oncogenesis, as it often results in the deregulated expression of the viral oncogenes E6 and E7, leading to inactivation of the tumor suppressor proteins p53 and pRb, increased genetic instability, and malignant transformation [14].

The persistence of high-risk HPV types, particularly HPV16 and HPV18, is strongly associated with the development of cervical and other anogenital cancers [12]. Understanding the complex interplay between HPV replication, immune evasion, and oncogenic potential is essential to develop effective prevention and treatment strategies.

## 4. HPV and Cancer

HPV is a significant etiological factor in various types of cancer, particularly cervical, anogenital (anal, vulvar, vaginal, penile), and head and neck cancers (oropharyngeal). Among cancer-causing HPV types, the most common and researched types are HPV-16 and HPV-18, which are responsible for more than 80% of cervical cancers worldwide, while other types are HPV-31, HPV-33, HPV-45 and HPV-58, of which HPV-16 is the most common type [16].

The process of HPV-induced carcinogenesis involves a complex network of interactions between viral oncoproteins and host cellular pathways, resulting in genomic instability, deregulation of cell cycle control, and evasion of apoptosis. The primary drivers of these processes are the viral oncoproteins E5, E6, and E7.

HPV-associated lesions often become cancerous through damage to the viral genome resulting from integration into the host genome. HPV oncoproteins E1 and E2 are involved in initiation and regulation of HPV infection, E4 protein is mainly involved in viral release, transmission and post-translational modification, E5, E6 and E7 in cellular proliferation, invasion and metastasis, cell cycle arrest, angiogenesis, resistance to cell death, tumor-promoting inflammation, genomic instability, evasion of growth suppressors, dysregulation of cellular energy, escape from immune destruction and replicative immortality [17]. The E6 protein is known to promote the degradation of the tumor suppressor protein p53, whilst the E7 protein is capable of inactivating the retinoblastoma protein (pRb), a process that leads to uncontrolled cell proliferation and the avoidance of apoptosis. The E5 protein has been demonstrated to increase the activity of both the E6 and E7 proteins and is involved in the regulation of cell proliferation and apoptosis. These proteins have also been observed to contribute to the initiation and progression of malignancy by interacting with cellular signaling pathways, including PI3K/AKT, Wnt and Notch [7,18] (Table 1).

In addition to the abovementioned oncoproteins, the integration of HPV into the host genome has been demonstrated to result in the increased expression of oncogenes. This process has been shown to induce the disruption of normal chromatin interactions and to silence tumor suppressor genes, a critical step in the process of carcinogenesis that leads to genomic instability [9]. Proteases such as matrix metalloproteinases (MMPs) have been observed to be upregulated in HPV-mediated carcinogenesis. The function of these enzymes is to facilitate tumor invasion and metastasis by degrading components of the extracellular matrix. Protease inhibitors have been identified as potential prognostic markers for HPV-associated cancers, emphasizing their role in disease progression [7].

With HPV integration, the expression of E6 and E7 oncogenes is dysregulated, and carcinogenesis is initiated, but this integration is not essential [13]. HPV is responsible for more than 90% cervical cancers, and more than 70% of these are related to HPV16 and HPV18. While HPV 18 and HPV48 show their carcinogenic effects by integrating into the host DNA, HPV 16 integrates at a rate of 60–80%. In approximately 30% of HPV16-positive HPV-related cancers, HPV 16 does not integrate into the host DNA and leads to carcinogenesis by deletion and rearrangement [19].

The pathogenesis of HPV-related squamous cell carcinoma and adenocarcinoma of the cervix involves different mechanisms, although both are driven by HPV infection. These differences can be explained by differences in the DNA integration point of HPV (8q24.21 is integrated in adenocarcinoma, while 21p11.2 is integrated in squamous carcinoma), the involvement of different genomic areas (dysregulation of oncogenes such as STARD3 and ERBB2 is observed in adenocarcinoma, while expression of viral oncogenes is increased in aquatic carcinoma), and different molecular signatures (squamous carcinoma is associated with keratinization and glucose metabolism pathways, while adenocarcinoma uses different oncogenic pathways) [14,20].

In HPV-related cancers, a different mechanism operates compared to HPV-independent cancers of the same region. This has resulted in HPV-positive cancers having better prognosis; for example, in cervical cancer, HPV positivity increases radiosensitivity [16]. HPV-related cancers are more commonly associated with squamous-type cancer, whereas HPV-independent tumors are associated with both adenocarcinomas and squamous histological subtypes; they are associated with earlier-stage lymph node involvement, more distant metastasis, and generally worse oncologic outcomes [21].

## 5. HPV Detection

HPV screening tests are an important health screening that should be performed in individuals with certain risk factors, especially due to the known cancer-risk-increasing effects of HPV. HPV infections are mostly asymptomatic and in most cases are cleared spontaneously by the immune system. However, some high-risk HPV types (especially HPV 16 and HPV 18) can cause the development of cervical, anal and other types of cancer. Therefore, HPV screening can help detect cancer at an early stage. High-risk groups such as women, HIV-positive individuals, men who have anal sex and those with weakened immune systems are the people who would benefit most from HPV screening tests.

The known tests for HPV diagnosis today are cervical screening and diagnostic tests. Tests for other body parts are still under development. Oral HPV tests and anal HPV tests are known tests, and genital tests for men are also under development. HPV DNA screening tests can be performed with swab samples, liquid-based cytology or tissue samples taken from areas such as the perianal region, oropharyngeal region, or skin. HPV is not a hematological virus; therefore, it cannot be screened using serum samples. However, there are studies that utilize antibodies, circular DNA, or fragments produced by HPV for screening purposes.

HPV is a virus that does not grow in cell culture and therefore requires molecular tests for diagnosis. HPV tests can be classified into three categories for cervical smear tests: (1) nucleic acid amplification assays, (2) nucleic acid hybridization assays, (3) signal amplification assays (Table 2) [22].

The recommended test for the diagnosis of HPV-associated oropharyngeal squamous cancer in the College of American Pathologists (CAP) and American Society of Clinical Oncology (ASCO) guidelines is p16 immunohistochemistry [23].

Anal area screening tests include a smear test and HPV DNA screening, just as in the cervical area. In addition, biopsies taken from HPV-induced lesions such as warts can also be diagnosed by HPV DNA screening.

It has been established that in numerous forms of cancer, DNA fragments, such as circulating tumor DNA (ctDNA), are released into the blood. In HPV-associated cancers, human papillomavirus (HPV) DNA can also be released into the blood following integration into the host genome. Consequently, the presence of circulating HPV-DNA fragments has the potential to serve as a marker for these cancers [24].

Next-generation sequencing (NGS) offers a non-invasive approach for the detection of circulating tumor DNA (ctDNA) in HPV-associated cancers, offering significant advantages over traditional methods such as droplet digital PCR (ddPCR) and quantitative real-time PCR (qPCR), particularly in terms of sensitivity, specificity and the ability to provide comprehensive genomic information, providing a non-invasive approach for diagnosis, monitoring and prognosis.

## 6. HPV and Geographic–Epidemiological Insights

The prevalence of HPV exhibits significant variation across geographic and demographic contexts, thereby influencing its correlation with upper gastrointestinal cancers and other HPV-associated malignancies. This variation is crucial for understanding the broader implications of HPV in cancer epidemiology.

The prevalence of HPV was found to be 11.7% worldwide, in a meta-analysis conducted by Bruni et al., which included 194 studies and more than 1 million women [22]. HPV is associated with 4.5% of all cancers worldwide and is responsible for 8.6% of cancers in women and 0.8% in men [25].

It is reported that 90% of cervical and anal cancers, 70% of vulvar and vaginal cancers, 60% of penile cancers and 70% of oropharyngeal cancers are attributable to HPV infection [26,27]. Approximately 60% of occurrences of oropharyngeal cancer (OPC) in the United States have been associated with HPV, compared to 31% in Europe and 4% in Brazil [28]. In contrast, India reports a high incidence of 38.4% of HPV-related OPCs, in contrast to the low rates observed in many African countries [29].

HPV represents the most significant etiological agent in the development of cervical cancer in women; nevertheless, it is also associated with anal and oropharyngeal cancers in men, and HPV-associated cancer rates are elevated in younger age groups, especially in areas exhibiting low vaccination rates [30].

The development of gastrointestinal cancers is often multifactorial, and HPV is one of several risk factors; in particular, the association of HPV with esophageal squamous cell carcinoma is very strong, but the evidence is insufficient for gastric or colorectal cancer [31]. Research indicates that the prevalence of HPV is elevated in patients diagnosed with oropharyngeal and gastrointestinal cancers, thereby suggesting a potential association between HPV 16 and 18 and the development of gastrointestinal system oncogenesis [31].

## 7. HPV and Esophagus

Esophageal cancer is the eighth most common cancer and the sixth most common cause of death [32]. There are two main histologic subtypes of esophageal cancer: squamous cell carcinoma (ESCC), 88% of cases, and adenocarcinoma (EAC), 12% of cases [33]. The most commonly known etiologies of ESCC are drinking, smoking, heredity, hot drinks, HPV infection and achalasia [34]. Recent studies have begun to investigate the mechanisms by which specific high-risk HPV types, particularly HPV 16 and HPV 18, may contribute to the development of esophageal cancer, leading to novel preventive strategies and therapeutic approaches targeting HPV.

HPV is a virus transmitted through contact and reaches the esophagus through orosexual contact. The relationship between HPV and ESCC was first demonstrated by Syrjänen in 1982, and as a result of the studies that followed, it is now known that the most common types are HPV 16 and HPV 18 [35,36]. HPV-associated ESCC of the esophagus occurs between 0 and 78% depending on geography, study design and method of HPV detection, and in regions with high incidence of ESCC such as Iran and Northern China, HPV incidence ranges between 32.8 and 63.6%, while in Europe and the USA where there is low incidence of ESCC, HPV incidence is 15.6% and 16.6% [33].

The results of a meta-analysis and review conducted by Petrelli et al. indicate that the association between HPV and ESCC is not as strong as that observed in cervical and oropharyngeal cancers [37]. Furthermore, there is no definitive correlation between HPV and p53 expression, a weak association with p16, and evidence suggests that ESCC environmental factors may be a more significant risk factor [33,37]. In a study reported by Feng et al. on Asian races, the oncogenic HPV E6 and E7 genes were shown to be involved in the pathogenesis of ESCC by upregulating susceptible HLA-DQB1 by DNA demethylation [38]. HPV-16 E6 contributes to EC carcinogenesis by downregulating microRNA-125b, a negative regulator of the Wnt/β-catenin signaling pathway [39]. Zhang et al. reported that HPV-infected ESCC tumor tissue, compared with HPV-negative tissue, contained longer telomeres due to the DNA methylation status of telomerase reverse transcriptase, indicating a relationship with poor prognosis [40].

Barrett’s esophagus (BE) is a metaplastic change in the esophageal epithelium, whereby the squamous cells are replaced by columnar cells. BE is the greatest risk factor for esophageal adenocarcinoma (EAC). The risk factors for BE include gastroesophageal reflux disease, smoking, obesity, hiatal hernia, male gender, white, and the sixth to seventh decade of life, and the risk factors for EAC include BE, hiatal hernia, and smoking [41]. Barrett’s esophagus affects about 5% of people in the US and about 1% worldwide, and first-line treatment consists of proton pump inhibitors for control of reflux symptoms [42]. There are some studies associated with HPV infection in EAC and BE, and it has been suggested that HPV may cause EAC and BE due to abnormalities in the p53 and retinoblastoma protein pathways, but these studies are very limited [1]. In a 2017 French study, 180 patients were studied, 61 of whom had BE, and no association between BE and HPV was found [43]. Although there are some studies showing a negative association between HPV and EAC and BE, the quality of the studies, geography, race and inappropriate tissue samples may have contributed to this result, but it is now recognized that approximately 25% of patients with EAC and BE are associated with HPV 16 and 18 [33]. The majority of HPV-positive BE and EAC samples show downregulation of pRb due to cleavage of pRb by the E7 oncoprotein, upregulation of p16INK4a, and overexpression of p53 due to inactivation by the E6 oncoprotein [44].

## 8. HPV and Stomach

Gastric cancer (GC) is the fifth most common type of cancer and the fourth most common cause of cancer-related death [45]. GC has a complex etiology that is influenced by a multifactorial genetic and environmental basis. The most well-established risk factors include smoking, alcohol consumption, family history, dietary habits, infection with the Epstein–Barr virus (EBV), and Helicobacter pylori (H. pylori). HPV is recognized for its association with cervical cancer; however, recent research suggests a potential link between HPV infection and gastric cancer. Understanding the biological pathways through which HPV may contribute to gastric carcinogenesis is essential to uncover potential preventive strategies and therapeutic targets.

In 2007, the International Agency for Research on Cancer analyzed and discussed previous studies and could not suggest an association between HPV infection and GC [46]. Studies on the association between HPV and GC are contradictory. It is considered that these differences are related to factors such as the year in which the studies were conducted, the design, the materials used and the test methods. In general, studies conducted in China show that there is a relationship between HPV and GC, and as a result of large meta-analyses conducted in recent years, it is seen that there is a positive relationship between HPV and GC, especially HPV 16 [33,47,48]. HPV positivity in gastric cardia cancers varies between 0 and 68% with the influence of factors such as geography, ethnicity and gender, and it is known that the HPV association is stronger than in non-cardia gastric cancers [49]. Although the presence of HPV in gastric cancer tissues has been shown in studies, the molecular relationship between HPV and carcinogenesis has not yet been clearly demonstrated.

Researchers have also examined the relationship between H. pylori, which is a known etiological factor of gastric cancer, and HPV. Studies have demonstrated a correlation between HPV positivity and H. pylori positivity in gastric cancer, with one potential interaction pathway being correlation of HPV 16 and H. pylori Cag A positivity, the development of dysplasia and adenocarcinoma due to HPV chronic infection, or HPV and H. pylori coinfection [5,50]. EBV is the second most important viral factor in the etiology of GC and accounts for approximately 10% of all gastric carcinomas. Similarly to EBV infection, HPV may play a role in carcinogenesis by stimulating the NFκB signaling pathway, which is critical for the proliferation and survival of cancer cells [51]. It has been established that HPV infection accelerates the carcinogenic process through various epigenetic mechanisms, including DNA methylation, histone modifications and changes in microRNA expression. HPV oncoproteins E6 and E7, for instance, have been demonstrated to promote a positive feedback loop by inhibiting histone acetyl transferase p300, thereby increasing their own expression and further contributing to malignancy through the process of epigenetic alteration [52].

A meta-analysis demonstrated an association between HPV 16 and gastric cancer (GC), with a reported prevalence of HPV of 23.6%. Although HPV screening with serologic tests is a reliable marker, the relationship between GC and HPV is more clearly evident in gastric tissue samples. It was hypothesized that HPV may play a role in oncogenesis by infecting gastric epithelial cells through oral entry, particularly by evading immune system mechanisms [47].

## 9. HPV and Duodenum

Small bowel cancers constitute 3% of all gastrointestinal system cancers [53]. Duodenal adenocarcinomas account for only 0.5% of all gastrointestinal tract cancers [54]. The molecular abnormalities observed in small bowel adenocarcinomas are prevalent in colon adenocarcinomas, but some occur with varying rates. There are no clearly defined environmental risk factors for small bowel adenocarcinomas, but approximately 20% are associated with predisposing diseases such as Crohn’s disease, Lynch syndrome, familial adenomatous polyposis, Peutz–Jeghers syndrome, and celiac disease [55]. There are studies showing that HPV is effective in colorectal carcinogenesis by inactivating p53 via the E6 oncoprotein. Considering the similarity of the molecular pathogenesis of small bowel cancers to the molecular pathogenesis of colorectal cancers, HPV may also be effective in small bowel cancers, but there is no study in the literature showing this relationship [51]. A study showing a relationship between HPV and duodenal cancer has not been seen in previous studies. However, it has been observed that cervical cancer and head and neck cancer may involve the duodenum through metastasis [56,57]. However, the association of HPV with cancer and metastasis in these patients has not been established.

## 10. Vaccine Status

HPV can cause cervical, vaginal, vulvar, penile, anal or oropharyngeal cancers, usually clearing within 2 years, but with the development of persistent infection, especially in types with a high risk of cancerization, such as HPV 16. HPV vaccines target the HPV types that cause most HPV-associated cancers, and studies have shown that vaccines are highly effective in preventing HPV-associated precancerous lesions and cancers [58].

It is recommended that HPV vaccines be administered at the age of 11–12 years. These vaccines have demonstrated strong efficacy against anogenital diseases and associated cancers, and they also have the advantage of potentially halting disease progression by preventing the spread of HPV in individuals being treated for HPV-related diseases [59].

HPV vaccines comprise three categories, all of which utilize DNA recombinant technology and L1 protein purification, a process which has been demonstrated to activate the immune system (Table 3) [60,61]. HPV vaccines are highly immunogenic, producing high concentrations of neutralizing antibodies against the antigen HPV L1 protein and activating both humoral and cellular immune responses. Numerous randomized clinical trials have demonstrated almost 100% efficacy in preventing HPV subtype-specific precancerous cervical cell changes, but vaccination does not provide protection in women already infected with HPV-16 or HPV-18 [62,63].

The increase in HPV-associated head and neck cancers, along with the demonstration of efficacy in different cancer types, such as ESCC, underscores the importance of HPV vaccination. The incidence of HPV-associated ESCC may be reduced through prophylactic HPV vaccination.

## 11. HPV-Related Treatment

HPV-related cancers may create differences and some advantages in treatment due to viral pathogenesis. In HPV-associated cervical cancers, when HPV is detected by a smear test to prevent the disease, cancer development can be prevented with minimal surgical treatments such as conization and cauterization in the precancerous period. Cervical, oropharyngeal and vulvar cancers are more radiosensitive when HPV-associated than when not, and this facilitates treatment [64]. In addition, preclinical and clinical studies for the treatment of HPV viral oncogenesis are ongoing. TALENs and CRISPR/Cas9 gene editing techniques, which aim to disrupt viral oncogenes by targeting E6 and E7 oncoproteins, the most important oncoproteins in HPV-associated cancers, and thus inhibit tumor growth and progression, are being investigated [65]. Another promising preclinical study in the treatment of HPV-associated cancers aimed to induce CD8+ T cell responses and kill tumor cells with customized viral immunotherapy targeting E6 and E7 proteins [66]. Oncolytic viruses are genetically modified or naturally occurring viruses that aim to induce an immune response against the tumor and, to do so, selectively infect and kill cancer cells while sparing normal cells. In HPV-associated cancers such as cervical cancer and head and neck squamous cell carcinoma, oncolytic viruses can be used more effectively as it is easier to recognize cancerous cells due to the nature of the cancer. Oncolytic HSV, especially genetically modified versions such as T-01, have been shown to inhibit the growth of cancer cells in HPV-related cervical cancer models by restricting the spread of the virus to tumors, leading to tumor cell death and immune response activation [67].

## 12. Future Research Directions

The role of HPV in the oncogenesis of upper gastrointestinal system cancers remains to be fully substantiated; however, there exist significant data indicating a correlation between the two. The multifaceted interactions among genetic, environmental, and viral factors in cancer development hinder a definitive evaluation of this relationship. In order to understand the pathogenesis and oncogenesis of HPV and to reveal its relationship with upper gastrointestinal system cancers, disciplines such as pathology, virology, oncology, surgery and gastroenterology should work together and create a common perspective.

Current prophylactic vaccines target specific high-risk HPV types but do not cover all oncogenic strains. Research is ongoing to develop vaccines that provide broader protection. One promising approach involves the use of the minor capsid protein L2, which is highly conserved across multiple HPV genotypes. Improving the immunogenicity of L2 by linking short amino acid sequences from different oncogenic HPV types or by displaying L2 peptides on more immunogenic carriers may lead to pan-HPV vaccines. Additionally, there is considerable interest in therapeutic vaccines aimed at eliciting immune responses against established HPV infections and associated malignancies. These vaccines primarily target the E6 and E7 oncoproteins that are consistently expressed in HPV-associated cancers. A variety of platforms are being investigated in clinical trials, including protein-based, viral vector, bacterial vector, and lipid-encapsulated mRNA vaccines [68].

One of the most challenging issues in HPV-related diseases is the lack of standardization in HPV diagnosis. In addition to tissue screening, the development of new diagnostic and screening tests from specimens such as blood and urine will pave the way for improvements in the diagnosis and treatment of HPV and related cancers. Early detection of HPV-associated lesions is crucial for effective intervention. New technologies such as DNA methylation triage, HPV integration detection, liquid biopsies, and AI-assisted diagnostics have the potential to augment traditional screening methods such as cytology and HPV nucleic acid testing. These innovations aim to increase sensitivity and specificity, reduce false-positive rates, and enable more personalized risk assessments. Further research is needed to validate and integrate these approaches into clinical practice [69].

Despite the proven efficacy of HPV vaccines, global immunization efforts face challenges such as low vaccination coverage, vaccine hesitancy, and inequalities in access to healthcare in low- and middle-income countries. Research focused on understanding the underlying causes of vaccine hesitancy, developing targeted education campaigns, and implementing policies to improve vaccine availability is critical. Additionally, exploring alternative vaccination strategies, such as single-dose regimens or needle-free delivery systems, may increase acceptance and coverage [70].

The heterogeneity of HPV-associated cancers necessitates personalized treatment approaches. Identification of biomarkers that predict response to treatment or disease progression may inform personalized treatment strategies. Research into the tumor microenvironment, immune response profiles, and genetic alterations associated with HPV-associated malignancies will aid in the development of personalized medicine approaches and potentially improve patient outcomes.

As HPV vaccination programs mature, long-term studies assessing the durability of vaccine-induced immunity and monitoring potential adverse events are essential. Such studies will inform booster vaccination programs when necessary and ensure continued safety and efficacy of vaccination programs.

## 13. Conclusions

The human papilloma virus (HPV) is one of the well-documented oncogenic viruses and has been the subject of extensive research. The fact that the source of cancer is a viral pathogen makes the disease both preventable through vaccination and treatable with antiviral therapy. Consequently, researchers are aiming to investigate viral diseases that may be causative agents in diseases such as cancer, which have poor prophylaxis, diagnosis, treatment, and prognosis. The objective is to make the disease less concerning by identifying these viral pathogens. The objective of this study was to synthesize the existing literature on the relationship between HPV and upper gastrointestinal tract cancers. We believe that further investigation into the relationship between HPV and ESCC, EAC, and GC is justified. Should the relationship and its underlying pathogenesis be elucidated through further study, it would have significant implications for the treatment of these cancers.

## Figures and Tables

**Table 1 viruses-17-00367-t001:** HPV proteins and functions.

Region	Protein	Pathway	Function
Early Region	E1	NF-κB	Genome replication
	E2		Regulation of gene transcription Leas to cancer progression
	E4		Viral release
	E5	MAPK-ERK pathways	Major oncoprotein E6 and E7 modulation Suppression of p21 expression Angiogenesis, proliferation, evading cell death
	E6	PI3K, AKT, Wnt, Notch pathways	Degradation of p53 Degradation of apoptotic signaling cascade Downregulates tumor suppressor genes Genomic instability, proliferation, invasion, immortality, inhibition of apoptosis and DNA repair
	E7	PI3K, AKT pathways	Inactivates pRb Inactivation of p21 and p27 Overexpression of MMP-9 Immortality, invasion, metastasis, chronic inflammation, genomic instability
	E8		Repressor of transcription
Late Region	L1		Major capsid protein
	L2		Minor capsid protein
Long Control Region			Promotor elements, regulator region

NF-κB: Nuclear Factor kappa B, MAPK-ERK: mitogen activated protein kinase- extracellular-signal-regulated kinase, PI3K: phosphatidylinositol-3-kinase, AKT: protein kinase B, DNA: deoxyribonucleic acid, MMP-9: matrix metalloproteinase-9.

**Table 2 viruses-17-00367-t002:** Classification of HPV tests for cervical specimens.

Classification of HPV Tests	
Nucleic Acid Amplification Assays	Micoarray PapilloCheck Polymerase Chain Reaction COBASE 4800 HPV Genome Sequencing CLART HPV2 INNO-LIPA
Nucleic Acid Hybridization Assays	Southern blot In situ hybridization Dot blot hybridization
Signal Amplification Assays	Cervista HPV Hyrid Capture 2

**Table 3 viruses-17-00367-t003:** HPV vaccine types.

Vaccine Type	HPV Type
Bivalent vaccine	HPV 16-18
Quadrivalent vaccine	HPV 9-11-16-18
Nonavalent vaccine	HPV9-11-16-18-33-35-4-52-58

## Data Availability

No new data were created or analyzed in this study. Data sharing is not applicable to this article.

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
