# Peer review of "Relationship Between Human Papilloma Virus and Upper Gastrointestinal Cancers"

_viruses, 2025, doi:10.3390/v17030367_

Round 1

Reviewer 1 Report

Comments and Suggestions for Authors

The manuscript examines the link between HPV and upper gastrointestinal cancers. It is organized into subsections for better comprehension of the topic.

The manuscript provides a detailed literature review on the HPV-cancer connection. Citing meta-analyses and epidemiological studies boosts its scientific validity.

Author Response

Manuscript Viruses-3419979B

Response to Reviewer 1

Dear Editor,

Thank you for giving us the opportunity to submit a revised draft of the manuscript “Relationship Between Human Papilloma Virus and Upper Gastrointestinal Cancers”  for publication in the Viruses. We appreciate the time and effort that you and the reviewers dedicated to providing feedback on our manuscript and are grateful for the insightful comments on and valuable improvements to our paper. We have incorporated most of the suggestions made by the reviewers. Please see below, in blue, for a point-by-point response to the reviewer 1 comments and concerns.

Reviewer  Comments to the Authors:

  1. The manuscript examines the link between HPV and upper gastrointestinal cancers. It is organized into subsections for better comprehension of the topic.

The manuscript provides a detailed literature review on the HPV-cancer connection. Citing meta-analyses and epidemiological studies boosts its scientific validity.

              Author response : Thank you for your valuable comments and suggestions. Your positive comments about our article were very valuable to us.

Sincerely,

19.02.2025

Ömer Vefik ÖZOZAN

Corresponding author on behalf of all authors

Reviewer 2 Report

Comments and Suggestions for Authors

See attachment

Author Response

Manuscript Viruses-3419979B

Response to Reviewer 2

Dear Editor,

Thank you for giving us the opportunity to submit a revised draft of the manuscript “Relationship Between Human Papilloma Virus and Upper Gastrointestinal Cancers”  for publication in the Viruses. We appreciate the time and effort that you and the reviewers dedicated to providing feedback on our manuscript and are grateful for the insightful comments on and valuable improvements to our paper. We have incorporated most of the suggestions made by the reviewers. Please see below, in blue, for a point-by-point response to the reviewer 2 comments and concerns.

Reviewer 2 Comments to the Authors:

Comment 1.  The question arises: Helicobacter pylory causes ulcers, that is, damage or directly oncogenic cell division. It may only cause an ulcer, but HPV is introduced to the damaged area, since the primary cellular defense is destroyed. That is, is it a combined infection like HIV and HPV?

Author response : Thank you for your valuable comments and suggestions. Research on the co-occurrence of Helicobacter pylori (H. pylori) and human papillomavirus (HPV) infections and their possible synergistic effects on cancer development is limited. The current literature does not clearly demonstrate the effects of co-infections of these two pathogens on gastric cancer. Therefore, more comprehensive and advanced research is needed on this subject.

Comment 2. Infection is only possible through injury; why does injury facilitate the passage of viral particles?

In general, the section is devoted to searching for the possibility of a combined effect of HPV and Helicobacter pilory, just as HIV inhibits the generation of CD4 T lymphocytes, reducing immunity and, as a result, facilitating HPV infection. But the authors do not indicate the root causes of this combined effect; it is possible that they simply were not discovered by nobody.

Author response : Thank you for your valuable comments and suggestions. As you noted, tissue damage weakens the body's natural barriers, making it easier for viruses to gain access to cells. Infections with Helicobacter pylori (H. pylori), in particular, disrupt these barriers, causing inflammation and ulcers in the gastric mucosa. This may allow other pathogens, such as human papillomavirus (HPV), to penetrate the damaged tissue more easily, increasing the risk of infection. However, current evidence on the synergistic effects of H. pylori and HPV co-infections on gastric cancer is limited and further research is needed.

Comment 3. The outer epidermis, consisting of more than two layers, is a stratified squamous epithelium consisting primarily of melanocytes and keratinocytes in the pharynx and upper zones of digestive tract. Squamocolumnar junction cells near transformed zone, are multipotent residual embryonic cells.

This type of erosion occurs in adolescence and can suddenly heal itself - true erosion is deformation of the stratified squamous epithelium.

Proliferation with dysplasia in stratified squamous epithelium is the most dangerous combination.

Koilocytes are squamous epithelial cells whose nucleus and cytoplasm have undergone degenerative changes. Their presence in a smear is a clear sign of HPV infection.

Stratified squamous epithelium is found in many places throughout the body. These locations include around various organs, the skin, the pharynx, and the esophagus.

According to GLOBOCAN, over the past 20 years, cancer mortality has increased by 200 cases per 100,000 people in Western Europe, North America, Canada and Australia. And this happened despite the fact that in these countries the population was completely vaccinated with the 2,4 and 9 valent preventive HPV vaccines. Still, the burden of cancer lies with HPV

Author response : Thank you for your valuable contributions. Your explanations about the biology of stratified squamous epithelium, erosion processes and HPV-related pathologies are very informative.

Comment 4. HPV viral particles can persist in stem cells, sperm, and even non-oncogenic cervical tissues. But under certain conditions: damage, loss of immunity and transplantation, HPV begins to replicate and rushes, as can be seen from the literature, to the sites of damage.

Which indicates to HPV particles that damage has occurred in the body and a situation has been created that facilitates the entry of the virus into human tissue? This question remains unexplored.

Author response : Thank you for your valuable comments and suggestions. Thank you for your valuable comments. The mechanisms of HPV latency and homing to sites of damage are still not fully understood. As you noted, the signals that signal HPV particles that damage has occurred have not been sufficiently investigated. Thank you for highlighting this important point; future studies should focus on better understanding these mechanisms.

Comment 5.Squamocolumnar junction cells near the zone of transformation appear to be multipotent embryonic cells in cervix are identical to gastroesophageal squamocolumnar junction cells. They are almost identical to anorectal squamocolumnar junction cells and HPV frequently determined in both types of cells in cervix and in anus. But the presence of HPV is not enough for transition from lesion to cancer.

Author response : Thank you for your valuable comments. We agree with your assessment regarding the similarity of squamocolumnar junction cells and the presence of HPV in these regions. However, we agree that the presence of HPV alone is not sufficient to initiate the cancer process. We would like to emphasize that immune response, genetic predisposition and environmental factors also play a role in this process. We will address these points in more detail in our study.

“HPV has an affinity for the squamocolumnar junctional tissue because basal cells are particularly accessible in the squamocolumnar transformation zone and are particularly susceptible to viral . Progressive acid damage to the esophagus may increase the likelihood of mucosal breaks that allow virus entry into the basal layer of the transformation zone. The SCJ is the transformation zone of the esophagus and resembles the transition zone of the uterine cervix, where nearly all high-grade cervical lesions and cancers arise. The presence of HPV in these areas alone is not sufficient for cancer development; cancer development is multifactorial.”

Reference:  Rajendra, P Sharma, Transforming human papillomavirus infection and the esophageal transformation zone: prime time for total excision/ablative therapy?, Diseases of the Esophagus, Volume 32, Issue 7, July 2019, doz008, https://doi.org/10.1093/dote/doz008

Comment 6-7-8. There is a very interesting hypothesis about the existence of a crystal lattice in the human body, because a person consists of 90-95% water. This crystalline grid can be defined as a hologram and was first used for Covid patients.

  1. It turned out that when someone gets sick with Covid, the hologram of the human body is distorted and even a diagnosis of Covid disease was created based on the distortions of this hologram. It follows that viruses can recognize changes in a person’s hologram or in his crystal lattice after damage to tissues or organs. And then it becomes clear how HPV can “see” the site of damage and rush to this damage in order to carry out an invasion and infect the person or another animal, revealing its hidden safe place where it manages to escape the human immune defense.
  2. Therefore, one can wonder for a long time why HPV carries the burden of cancer, why there are co-infections and why carcinogenesis is increasing despite all vaccination efforts

Author response : Thank you for your review. Although our study does not directly include the holographic model and crystal lattice hypothesis, the ability of HPV to infect damaged tissues can be explained by immune escape mechanisms and epithelial cell renewal. The increase in cancer burden and co-infections require comprehensive studies that take into account not only vaccination but also environmental and genetic factors.

Sincerely,

19.02.2025

Ömer Vefik ÖZOZAN

Corresponding author on behalf of all authors

Reviewer 3 Report

Comments and Suggestions for Authors

Human papillomavirus (HPV) is primarily associated with sexually transmitted diseases of the genital area, but is also responsible for diseases affecting the skin and mucosal areas.  Recently, there has also been an increasing awareness that HPV may play a role in cancers of the head and neck, as well as those of upper gastrointestinal tract.  This review focuses on the latter, examining the evidence for a causal relationship between HPV and different types of cancer of the upper GI tract, as well as the oncogenic mechanisms involved, the relative prevalence (including geographically speaking) and the current and future prospects for treatment.

The authors present a systematic evaluation of all aspects of the topic of HPV and its role in upper GI cancers.  They begin with a concise, but comprehensive, summary of the HPV virus genome and its encoded proteins.  This is followed by a discussion of the most important types of HPV, in the process noting those that are the most closely associated with various types of cancers.  Then, they summarize the life cycle of the virus with a focus on its interaction with specific cell types, including its ability to integrate into the host DNA, as well as its ability to escape immune surveillance.  After a highly informative summary of the roles of the various viral proteins in the virus life cycle, they discuss in detail the mechanistic bases for HPV oncogenesis, especially involving the E6 and E7 proteins.  This section of the review is exquisitely written, making a highly complex process exceptionally clear and easy to follow.  As one comparatively unfamiliar with DNA virus oncogenesis, their attention to detail and logical presentation is much appreciated.

The in-depth discussion of what is known about the prevalence and mechanistic basis of oncogenesis in the esophagus, stomach and small bowel is considered appropriately balanced, recognizing the dearth of evidence for a role of HPV in duodenal cancers, after highlighting the known multifactorial genetic and environmental, including HPV, bases for the etiology of both esophageal and gastric cancers.  I found this portion of the review to be quite extremely well composed and quite interesting. 

The review ends with a summary of the current status of HPV vaccines and treatments, followed by a discussion of future research prospects, which I found quite encouraging, albeit in the face of ever-increasing vaccine hesitancy.

 This review mounts a cogent and compelling argument that HPV plays a significant role in the development of many upper GI cancers.  It is considered an outstanding treatise on the topic with no significant deficiencies noted.  Only one very minor point the authors may want to consider adding is that an effective treatment for Barrett’s esophagus is the use of proton pump inhibitors (PPIs) to reduce the production of stomach acid and relieve the symptoms of gastroesophageal reflux disease (GERD) and reduce damage to the lower esophagus that can lead to esophageal cancer.  Other than that very minor issue, the manuscript is recognized as an accurate and timely evaluation of the present state of the field.

Author Response

Manuscript Viruses-3419979B

Response to Reviewer 3

Dear Editor,

Thank you for giving us the opportunity to submit a revised draft of the manuscript “Relationship Between Human Papilloma Virus and Upper Gastrointestinal Cancers”  for publication in the Viruses. We appreciate the time and effort that you and the reviewers dedicated to providing feedback on our manuscript and are grateful for the insightful comments on and valuable improvements to our paper. We have incorporated most of the suggestions made by the reviewers. Please see below, in blue, for a point-by-point response to the reviewer 3 comments and concerns.

Reviewer 3 Comments to the Authors:

  1. Human papillomavirus (HPV) is primarily associated with sexually transmitted diseases of the genital area, but is also responsible for diseases affecting the skin and mucosal areas.  Recently, there has also been an increasing awareness that HPV may play a role in cancers of the head and neck, as well as those of upper gastrointestinal tract.  This review focuses on the latter, examining the evidence for a causal relationship between HPV and different types of cancer of the upper GI tract, as well as the oncogenic mechanisms involved, the relative prevalence (including geographically speaking) and the current and future prospects for treatment.

The authors present a systematic evaluation of all aspects of the topic of HPV and its role in upper GI cancers.  They begin with a concise, but comprehensive, summary of the HPV virus genome and its encoded proteins.  This is followed by a discussion of the most important types of HPV, in the process noting those that are the most closely associated with various types of cancers.  Then, they summarize the life cycle of the virus with a focus on its interaction with specific cell types, including its ability to integrate into the host DNA, as well as its ability to escape immune surveillance.  After a highly informative summary of the roles of the various viral proteins in the virus life cycle, they discuss in detail the mechanistic bases for HPV oncogenesis, especially involving the E6 and E7 proteins.  This section of the review is exquisitely written, making a highly complex process exceptionally clear and easy to follow.  As one comparatively unfamiliar with DNA virus oncogenesis, their attention to detail and logical presentation is much appreciated.

The in-depth discussion of what is known about the prevalence and mechanistic basis of oncogenesis in the esophagus, stomach and small bowel is considered appropriately balanced, recognizing the dearth of evidence for a role of HPV in duodenal cancers, after highlighting the known multifactorial genetic and environmental, including HPV, bases for the etiology of both esophageal and gastric cancers.  I found this portion of the review to be quite extremely well composed and quite interesting. 

The review ends with a summary of the current status of HPV vaccines and treatments, followed by a discussion of future research prospects, which I found quite encouraging, albeit in the face of ever-increasing vaccine hesitancy.

 This review mounts a cogent and compelling argument that HPV plays a significant role in the development of many upper GI cancers.  It is considered an outstanding treatise on the topic with no significant deficiencies noted.  Only one very minor point the authors may want to consider adding is that an effective treatment for Barrett’s esophagus is the use of proton pump inhibitors (PPIs) to reduce the production of stomach acid and relieve the symptoms of gastroesophageal reflux disease (GERD) and reduce damage to the lower esophagus that can lead to esophageal cancer.  Other than that very minor issue, the manuscript is recognized as an accurate and timely evaluation of the present state of the field.

              Author response : Thank you for your valuable comments and suggestions. Your positive comments about our article were very valuable to us. As you stated in your suggestion, a sentence about Barrett's esophagus treatment with proton pump inhibitors was added to the article. The correction is shown below.

“Barrett's esophagus affects about 5% of people in the US and about 1% worldwide, and first-line treatment consists of proton pump inhibitors for control of reflux symptoms.”

Referrance: Sharma P. Barrett Esophagus: A Review. JAMA. 2022;328(7):663–671. doi:10.1001/jama.2022.13298.

Sincerely,

19.02.2025

Ömer Vefik ÖZOZAN

Corresponding author on behalf of all authors